# Electromagnetic Interference Shield of Highly Thermal-Conducting, Light-Weight, and Flexible Electrospun Nylon 66 Nanofiber-Silver Multi-Layer Film

**DOI:** 10.3390/polym12081805

**Published:** 2020-08-11

**Authors:** Jaeyeon Kim, Suyeong Lee, Changho Kim, Yeongcheol Park, Mi-Hyun Kim, Jae Hun Seol

**Affiliations:** 1School of Mechanical Engineering, Gwangju Institute of Science and Technology (GIST), Buk-gu, Gwangju 61005, Korea; bestkjy0426@gmail.com (J.K.); silver4fox@gm.gist.ac.kr (S.L.); umtoss530@gist.ac.kr (C.K.); young13id@gist.ac.kr (Y.P.); 2ICT Materials & Components Research Laboratory, ETRI, 218 Gajeong-ro, Yuseong-gu, Daejeon 305700, Korea; kmhyun@etri.re.kr

**Keywords:** electrospun nylon 66 nanofiber, nanofiber-Ag multi-layer composite, electromagnetic interference shielding, heat dissipation, thermal conductivity

## Abstract

A light-weight, flexible electromagnetic interference (EMI) shield was prepared by creating a layer-structured metal-polymer composite film consisting of electrospun nylon 66 nanofibers with silver films. The EMI shielding effectiveness (SE), specific SE, and absolute SE of the composite were as high as 60.6 dB, 67.9 dB cm^3^/g, and 6792 dB cm^2^/g in the X- and K_u_-bands, respectively. Numerical and analytical calculations suggest that the energy of EM waves is predominantly absorbed by inter-layer multiple reflections. Because the absorbed EM energy is dissipated as heat, the thermal conductivity of absorption-dominant EMI shields is highly significant. Measured thermal conductivity of the composite was found to be 4.17 Wm^−1^K^−1^ at room temperature, which is higher than that of bulk nylon 66 by a factor of 16.7. The morphology and crystallinity of the composite were examined using scanning electron microscopy and differential scanning calorimetry, respectively. The enhancement of thermal conductivity was attributed to an increase in crystallinity of the nanofibers, which occurred during the electrospinning and subsequent hot pressing, and to the high thermal conductivity of the deposited silver films. The contribution of each fabrication process to the increase in thermal conductivity was investigated by measuring the thermal conductivity values after each fabrication process.

## 1. Introduction

As electronic and telecommunication devices have become faster and now consume more power, electromagnetic radiation, which is emitted from these devices, causes undesirable and sometimes harmful effects. These involve not only the performance of electronic devices (i.e., electromagnetic interference: EMI) [1,2], but also induce effects in human bodies [3,4,5]. The frequency range involving EMI can be extremely broad in a variety of applications [6,7,8,9]. Electromagnetic interference shielding of high frequencies has attracted substantial interest since the advent of wireless internet connection and mobile devices (e.g., 5G cellular phones operate above 6 GHz), wireless local area networks (LAN, 2.4 GHz, and 5 GHz) [10], and the internet of things (IOT) [11], in addition to the traditional demands for military and aerospace applications [12,13]. As for EMI shielding in such mobile and wireless internet devices, the weight, flexibility, and processability of EMI shielding materials should be considered, as well as their shielding effectiveness (SE).

Although metals have been widely used because of their high electrical conductivity [14,15,16], they have disadvantages such as heaviness, inflexibility, and low chemical stability. Alternatively, polymers have risen as substitutes for metals when they are mixed with electrically conducting or magnetic materials (e.g., carbon fibers [17,18], carbon nanotubes (CNT) [19,20], graphene [21,22], and metal nanowires (NW) [23,24,25]). However, strong reflections from the surfaces of materials that are highly conductive electrically, tend to yield secondary pollution, which leads to malfunction of neighboring electronic devices [26]. Thus, absorption-dominant EMI shielding materials have been developed, which include, for example, polymer-magnetic nanoparticles [27], hydrophobic-treated polymer-metallic nanoparticle [28], and polymer-graphene composites [29,30]. Additionally, several recent studies have reported that metal-coated electrospun nanofibers (NFs) work as EMI shielding materials. This results from multiple-reflections between the metal-coated NFs and an increase in the effective surface areas [31,32]. The nanofibrous structure with high porosity not only induces other internal multiple reflections and dielectric loss as an absorption mechanism [33], but also achieves high specific shielding effectiveness (SSE). The latter is obtained by dividing an EMI SE into its density [34]. When absorption plays a dominant role for EMI shielding, the absorbed energy of an EM wave is transformed into thermal energy, which could increase the temperature of an absorbing medium, e.g., a polymer composite, excessively. Thus, it is extremely important to dissipate the absorbed thermal energy effectively by increasing the thermal conductivity of the polymer composite. Otherwise, the inherently low thermal conductivity of the polymer composite may result in an unwanted excessive temperature increase and consequently degrade the performance and long-term durability of shielded devices. To address this issue, there have been studies aimed at dissipating the heat energy generated by adding fillers with high thermal conductivity [35,36,37,38] or by depositing metal films [39,40].

In the current study, silver (Ag)-deposited electrospun nylon 66 nanofibrous mats and non-porous nylon 66 films were fabricated. In addition to the EMI SE of the mat and film, we first report the thermal conductivity of these mats and the films. Although the contribution of the Ag layer to the overall thermal conductivity of the mats was not taken into account, the thermal conductivity of the mats was thought to be significantly higher than that of their bulk counterparts in view of the increases in thermal conductivities of individual electrospun polymer NFs. These were induced by enhanced molecular chain alignment or crystallinity [41,42]. Therefore, the thermal conductivity values of individual electrospun nylon 66 NFs were measured with suspended microdevices. Regarding the EMI SE, it was found that the orientation of the NFs, which was determined during the electrospinning processes, affected the EMI SE of the mats when the electromagnetic waves were polarized. In addition, the EMI absorption mechanism in the mats and films was analyzed using numerical and analytical calculations.

## 2. Materials and Methods

### 2.1. Materials

Nylon 66 pellets (molecular weight = 262.35 g/mol) and formic acid (reagent grade, ≥95%) were supplied from Sigma-Aldrich, St. Louis, MO, USA. A silver source (99.99%, granules 3–5 mm) for electron beam evaporation was purchased from Taewon Scientific Co., Ltd., Seoul, Korea.

### 2.2. Preparation of Electrospun Nylon 66 Mat and Non-Porous Nylon 66 Film

Nylon 66 pellets were used for the synthesis of electrospun NFs. The nylon 66 solution was made by dissolving the nylon 66 pellets in formic acid at 40 °C with stirring for 3 h. Two nylon 66 solutions with different weight percent (7.5 and 15 wt%) were prepared for fabricating electrospun nylon 66 nanofibrous mats and non-porous nylon 66 films, respectively.

To fabricate an electrospun nylon 66 nanofibrous mat (hereafter referred to as electrospun mat), an electrospinning apparatus (Nano NC Co., ESR100D, Seoul, Korea) was used under the following conditions: an applied voltage of 21 kV, the pumping rate of the syringe of 3.5 μL/min, and distance between the syringe needle and the drum collector of 20 cm. Based on previous studies [43,44], the rotation of the drum collector, of which the rotation speed was 1600 rpm, was adopted in order to reduce the randomness of the NF orientation in the mat. At a spinning speed above 1600 rpm, it was difficult to keep the thickness of the electrospun mats uniform. Under these conditions, a 20-μm-thick electrospun mat was fabricated with 4 mL of nylon 66 solution. Afterward, the electrospun mats were ethanol-treated in order to improve adhesion between the mats because they inherently repulse each other [45]. Each electrospun mat was immersed in ethanol (80 *v*/*v*) on a hot plate, of which the temperature was maintained at 70 °C for an hour. The ratio between the volume (mL) of the treatment ethanol to the weight (g) of the electrospun mats was 80:1. Afterward, the ethanol-treated electrospun mats were rinsed with deionized water and dried in an oven at 50 °C, for 2 h. For handling convenience, three layers of electrospun mats were hot-pressed into a single layer, which is referred to as a hot-pressed electrospun mat hereafter, at a temperature of 120 °C under the pressure of 10 MPa for 50 min. The electrospun mats were stacked along with the orientation of the NFs in the mats before the hot-pressing process. The thickness of the hot-pressed electrospun mats was regulated by inserting 30-μm-thick feeler gauges between the two pressing plates. To make the non-porous nylon 66 films (hereafter referred to as non-porous films), the 15 wt% nylon 66 solution was poured on a flat metal plate and spread using a doctor blade (IMOTO Machinery Co., Ltd., IMOTO 1117-200, Kyoto, Japan). Subsequently, the films were dried at room temperature for an hour. Similar to the electrospun mat, the non-porous films were ethanol-treated before they were hot-pressed.

### 2.3. Fabrication of Multi-Layered Nylon 66-Ag for an EMI Shield

As listed in Table 1, there were four types of samples, which were classified according to their structures (i.e., nanofibrous mat or non-porous film) and to the presence (or absence) of Ag layers. The five hot-pressed electrospun mats were piled up with NFs in the same axial direction in each mat and subsequently hot-pressed. The mat with this arrangement was labeled E-0. On the other hand, for N-0, five non-porous films were stacked and subsequently hot-pressed. Both E-0 and N-0 were hot-pressed at 120 °C under a pressure of 10 MPa for 50 min. To fabricate EMI shielding composites, 50-nm-thick Ag layers were deposited on hot-pressed electrospun mats and non-porous films, respectively, using an electron beam evaporator (SNTek, REP5004, Gyeonggi-do, Korea) with a deposition rate of 0.7 Å/s. The thickness of the Ag layer was set in the range 30–130 nm, which was determined by taking into account the desire to maintain the optical properties of the bulk and to form pores on the Ag layer, respectively [46]. The electrospun mats, with or without the Ag-deposited layers, were piled up with the representative axes of the NFs aligned and subsequently hot-pressed at 120 °C into a single, merged electrospun mat, denoted by E-50. The same fabrication process, which was applied to produce non-porous films instead of electrospun mats, produced a merged film, i.e., N-50. Figure 1a,b show multi-layer structures of E-50 and N-50, which consist of the five hot-pressed electrospun mats with four Ag-deposited layers between them. Appendix A) shows the morphology of the Ag layers deposited on the non-porous film. For fabricating N-50, the hot-pressing temperature was increased to 160 °C to improve the adhesion between the Ag-deposited films. Each of the hot-pressing temperatures was chosen to prevent insufficient adhesion between the Ag-deposited mats and the non-porous films, as well as the degradation of their mechanical properties, which would occur at excessively low and high temperatures [47], respectively. The former and latter conditions indicate the lower and upper bounds of temperature in this process. The thickness of all the composites was controlled using 100-μm-thick feeler gauges inserted between the two hot pressing plates.

### 2.4. Characterization

#### 2.4.1. Morphology of the Electrospun Mats and Films

Scanning electron microscope (SEM, Hitachi S4700, Tokyo, Japan) images of the electrospun mats and films were obtained with an accelerating voltage of 10 kV. To enhance the resolution of the images, a less than 10-nm-thick platinum layer was deposited on the electrospun mats and films. SEM images were taken of each fabrication procedure (i.e., electrospinning, ethanol treatment, hot-pressing, or metal deposition). The average diameter, diameter distribution of NFs, and pore size distribution consisting of the electrospun mat or the Ag-deposited electrospun mat in the EMI composites were measured in five different SEM images of electrospun mats using Image J software (National Institutes of Health, Bethesda, MD, USA). The pore diameters were obtained under the assumption that the pores have a circular shape. Moreover, the porosity of the fabricated EMI shielding composites is determined by the relation between the true density and the bulk density. The former and the latter were measured using a gas pycnometer and based on mass and volume according to ISO 845, respectively.

#### 2.4.2. Thermal Conductivity Measurements of Individual Electrospun Nylon 66 NFs, Mats, and Films

As shown in Figure 2, the thermal conductivity of individual electrospun nylon 66 NFs with various diameters, was measured using suspended micro-devices developed for measuring the thermal conductivity of one-dimensional materials with high sensitivity [48,49]. Under the same electrospinning conditions applied to the fabrication of the electrospun mats, the nylon 66 NFs were electrospun directly onto the suspended micro-devices attached to the grounded drum collector. Except for an individual NF, of which the thermal conductivity values were measured, all the other redundant NFs on the suspended micro-devices were removed using a micro-manipulator. Furthermore, both extruding parts of the NF to be measured beyond the membranes were cut off using an argon-ion laser. The latter was installed in a Raman spectrometer (Renishaw, Renishaw inVia Raman microscope, Gloucestershire, UK) with a power of 10 mW. The thermal conductivity of the individual NFs was measured in a cryostat environment (JANIS, closed-cycle CCS-400/204 helium cryostat, Woburn, MA, USA) over a temperature range of 100 to 300 K. The thermal conductivity of the mats/films was obtained based on the following definition of thermal diffusivity (i.e., $k=\alpha \times \rho \times C_{p}$ [50]), where *k*, *α*, *ρ*, and *C_p_* are the thermal conductivity, the thermal diffusivity, the true density, and the specific heat capacity, respectively. Each parameter (*α*, *ρ*, and *C_p_*) was measured using the laser flash method, a gas pycnometer, and differential scanning calorimetry (DSC), respectively.

#### 2.4.3. Differential Scanning Calorimetry (DSC)

The crystallinity of the electrospun mats and nonporous films was characterized by a non-isothermal differential scanning calorimeter (TA Instrument Inc., Q20, New Castle, DE, USA). The nylon 66 mats and films, of which the weights were approximately 5–10 mg, were prepared for the DSC characterization, and the measurements were carried out in a temperature range of 30 to 300 °C with a heating rate of 10 °C/min and nitrogen purging.

#### 2.4.4. Electromagnetic Interference Shielding Effectiveness (EMI SE)

The electromagnetic interference SE of the mats/films was characterized using a network analyzer (KEYSIGHT, E5063A, Santa Rosa, CA, USA) in frequency ranges of the X-band (8.2–12.4 GHz) and the K_u_-band (12.4–18.0 GHz). Scattering parameters (i.e., S_11_ and S_21_) were obtained from transverse-electric TE_10_ modes, which were supported by rectangular waveguides according to the waveguide standard: WR 90 and WR 62 for the X- and K_u_-bands, respectively. Total (SE_T_), reflection (SE_R_), and absorption (SE_A_) SE values were calculated based on the measured scattering parameters and using the following equations [51].
(1)$${\mathrm{SE}}_{R}=-10\;\log\left(1-{\left|S_{11}\right|}^{2}\right)$$
(2)$${\mathrm{SE}}_{R}=-10\;\log\left(1-\frac{{\left|S_{21}\right|}^{2}}{1-{\left|S_{11}\right|}^{2}}\right)$$
(3)$${\mathrm{SE}}_{T}={\mathrm{SE}}_{A}+{\mathrm{SE}}_{R}$$

#### 2.4.5. Electrical Conductivity of the Deposited Ag Layers

The electrical conductivities of the 50-nm-thick Ag layers deposited on the electrospun mat and the non-porous film were measured based on the equation $\sigma ={\left(R_{s}\cdot t\right)}^{-1}$, where σ, *R_S_*, and *t* are the electrical conductivity, the electrical sheet resistance, and the thickness of the deposited Ag, respectively. The electrical sheet resistance was measured using the four-point probe method with a multimeter (Agilent, 34401A, Santa Clara, CA, USA) and a probe station (Cascade Microtech, Beaverton, OR, USA). The electrical conductivities of the Ag-deposited electrospun mats were measured in directions both parallel and perpendicular to the orientation of the NFs in the mats.

#### 2.4.6. Mechanical Tests

To analyze the mechanical behavior of the EMI shielding composites, bending and static tensile tests were performed using a custom motorized stage and a precision universal tester (Shimadzu, Autograph AG-X, Kyoto, Japan) according to ASTM D882, respectively. The bending test was conducted to evaluate flexibility and attachment of the interfaces between the Ag and nylon 66 layers in E-50 and N-50. Each test was repeated for 3000 cycles. Specimens were prepared in the same way as mentioned above, and for the tensile test, the gauge length was 125 mm and was tested at 50 mm/min. A stress-strain curve is determined based on the measured force-strain curve and specimen cross-sectional area in SEM images. 

## 3. Results and Discussion

### 3.1. Morphology Analysis

The morphology of the fabricated nanofibrous mats was investigated with SEM to examine the effect of the fabrication process on the mat morphology. As shown in Figure 3a, the as-spun mat consists of well-aligned NFs without beads, which are separated from each other because of electrostatic repulsion force [52]. The diameter of the NFs ranged from 75 to 187.6 nm and the average diameter was 116.0 nm as shown in Figure 3e. The low hydrophilicity of nylon 66 NFs, which originates from methylene groups [53], leads to the low adhesion energy according to Young’s equation [54] as well as the large contact angle of the mats [55]. Because of such a lack of adhesion energy on the NFs, the as-spun mats adhered poorly to each other in spite of being hot-pressed. In a previous study [45], an aqueous ethanol solution was used to disorder the methylene groups of the nylon 66 NFs, resulting in increases in the adhesion energy and hydrophilicity of the NFs without changing their surface geometry. Therefore, ethanol-treatment was performed on the electrospun mats before the hot-pressing process. As shown in Figure 3a,b, the morphology of the ethanol-treated electrospun mat shows that the NFs became closer in comparison with those of an untreated mat. In Figure 3c, an ethanol-treated and subsequently hot-pressed electrospun mat has denser NFs owing to the reduction of the thickness by the hot pressing. Such densification can contribute to enhanced thermal conductivity in the hot-pressed electrospun mats (i.e., k∝ρ). Additionally, the average diameter of NFs was slightly increased by approximately 15 nm during the ethanol treatment and hot-press processes as shown in Figure 3f. Figure 3d shows that the 50-nm-thick Ag layer deposited on the mat confirmed the nanofibrous structure of the electrospun mat because of its relative thinness (compared with the diameter of the NFs). In the Ag deposition process, the pore diameters were entirely reduced, and the average pore diameter was decreased by 41.4 nm (see Appendix A). The pores in the Ag layer can induce additional multiple reflections and dielectric loss [33].

### 3.2. Electrical Properties of the Composites

Electrical conductivity is one of the crucial factors that determines EMI shielding characteristics such as reflection, absorption, and multiple reflections [56]. Therefore, the electrical conductivities were measured using the four-point probe method as shown in Figure 4. All of the electrical conductivities are lower than that from bulk Ag, which is 6.14 × 10^7^ S/m at room temperature [57], because of the reduction of the electron mean free paths (MFPs) in the Ag layers. The average electron MFP in bulk Ag is 53.3 nm at room temperature [58]. However, electron MFPs may be significantly shortened in the 50-nm-thick Ag layers because of the reduction of grain size and grain boundary reflection, which lead to a decrease in electrical conductivity [59]. The electrical conductivity of N-50 was higher than that of E-50 by approximately a factor of two, even in the case that the NFs in E-50 were aligned with the electrical current flow for the four-point probe measurement. The reduction of the electrical conductivity in E-50 may be attributed to the discontinuity of the electrical paths due to the porous structure and to the non-uniform Ag thickness along the circular arc of the NF cross-section [60]. Although the electrical conductivity of N-50 is isotropic in the film plane, that of E-50 is anisotropic, depending on the angles between the currents of the four-point probe measurements and the orientations of the NFs. The electrical conductivity of E-50 with NFs paralleling the current was 4.9-fold higher at room temperature than that with the NFs perpendicular to it. As shown in Figure 3d, the anisotropic morphology of the Ag layer remains intact after the hot-pressing process, resulting in the anisotropy of electrical conductivity in E-50.

### 3.3. Thermal Properties of the Composites

Figure 5a shows the improved in-plane thermal conductivity values of E-0, E-50, N-0, and N-50 compared with that of bulk nylon 66, which was also reported in previous research [61]. Through the electrospinning, ethanol treatment, and hot-pressing processes, the thermal conductivity of E-0 was increased by a factor of 13.4, with respect to that of bulk nylon 66. Among these fabrication processes, electrospinning could enhance the thermal conductivity of a polymer NF via alignment of the molecular chains [62], and heat treatment in a hot-pressing process could contribute to an increase in the crystallinity of a polymer, which accordingly would increase the thermal conductivity [63,64]. In this study, the contribution of each fabrication process to the thermal conductivity was investigated separately.

As mentioned above, the thermal conductivity of individual NFs was measured using suspended micro-devices to evaluate the contribution of the electrospinning process. The measurement results in Figure 5b indicate that the measured thermal conductivity of individual NFs, of which the diameters range from 61.6 to 125.1 nm, was 3- to 5-fold higher than that of bulk (0.25 Wm^−1^K^−1^ at room temperature [61]. The upper limit diameter of the individual NFs was close to the average diameter of NFs in the as-spun mat. During the electrospinning process, molecular chains in the polymer were stretched by the electrostatic force between the syringe and the rotating drum, thereby increasing their thermal conductivity. As the diameter of an NF decreases, its thermal conductivity increases because the molecular chains become more aligned in the axial direction of the NFs [41]. As shown in Figure 5a, the thermal conductivity of E-0 is much higher than that of all individual NFs although the thermal conductivity of an as-spun mat is expected to be lower than that of an individual NF due to imperfection of the NF alignment. This result indicates that the increase in thermal conductivity of E-0 results from not only the electrospinning process but also the other fabrication processes (e.g., ethanol treatment and hot-pressing).

The crystallinity of nanofibrous mats after each fabrication stage was investigated because the thermal conductivity of semi-crystalline polymers is significantly affected by their crystallinity [63]. Corresponding to the fabrication stages, Figure 5c shows the degrees of crystallinity of three kinds of mats (as-spun, ethanol-treated, and ethanol-treated and subsequently hot-pressed at 120 °C), which were calculated from ratios between measured melting enthalpies ($\Delta H_{m})$ of the mats and that of perfect crystalline nylon 66 $\left(\Delta H_{m}^{0}=200.8\;J/g\right)$ [65]. Meanwhile, the crystallinity of bulk nylon 66 was obtained from the density of nylon 66 pellets (1.14 g/mL at 25 °C, Sigma Aldrich) and those of fully crystalline and amorphous nylon 66 [66], i.e., $\chi =\left(\rho_{p}-\rho_{a}\right)/\left(\rho_{c}-\rho_{a}\right)$, where *ρ_p_*, *ρ_a_*, and *ρ_c_* are the densities of the pellet, the amorphous phase, and the crystalline phase, respectively, and *χ* is the crystallinity. The degree of crystallinity was gradually elevated as the bulk nylon 66 pellets went through the fabrication stages. A relatively large increase in crystallinity by electrospinning was observed because the molecular chains became aligned and hydrogen bonds easily formed crystalline structures during electrospinning [67]. Additionally, when nylon 66 mats were rinsed in deionized water after the ethanol treatment, the mobility of the molecular chains was increased, leading to enhancement of the crystallinity [68]. Because of the comparatively low hot-pressing temperature of 120 °C, the crystallinity did not increase significantly after the hot-pressing process. This is contrary to the results in previous studies [66,69,70], which reported that an increase in the thermal conductivity resulted from crystallinity enhancement through rearrangement of the molecular chains and modification of the regularity of the molecular chains between major lamellae. As shown in Appendix A, the crystallinity was not increased noticeably after the hot-pressing process. Consequently, the degree of crystallinity of E-0 increased from that of bulk through each fabrication process in agreement with the increased thermal conductivity of E-0. Furthermore, the increase of thermal conductivity for E-0 may also originate from the change of the molecular orientation during fabrication processes, which does not involve crystallinity but does enhance thermal conductivity [41]. As a future study, it would be of great interest to characterize the molecular orientations of NFs during the fabrication processes using polarized Raman [71], X-ray diffraction, or polarized infrared spectroscopy [72] analyses.

In addition to the high thermal conductivity of E-0, the introduction of Ag into E-0 increases the thermal conductivity of E-0 by 23.8% as shown in Figure 5a. Such an increase in thermal conductivity is attributed to the thermal conductivity of the Ag layers. The thermal conductivity of bulk Ag is 429 Wm^−1^K^−1^ at room temperature [73]. To better understand this result, the contribution of the Ag layers to the thermal conductivity of E-50 was theoretically calculated based on the Wiedemann-Franz law:(4)$$k_{e}/\sigma =LT,$$
where *k_e_*, *σ*, *L*, and *T* represent the electronic thermal conductivity, electrical conductivity, Lorenz number, and temperature, respectively [74]. In a thin metal film, of which the thickness is less than that of electron MFPs, electrons are scattered, leading to the reduction of electrical and thermal conductivity. In view of the reduction of electron MFPs, the measured electrical conductivity of the 50-nm-thick Ag layer in E-50 was applied to the Wiedemann–Franz law together with the corresponding Lorenz number [75]. The result was thermal conductivity of 46 Wm^−1^K^−1^ at room temperature. Because the measured electrical conductivity of the Ag layer in E-50 parallel to the axis of its NFs was used for the calculation, the actual in-plane thermal conductivity of E-50 would be lower than the calculated value due to the lower electrical and thermal conductivities in the direction perpendicular to the NFs. Assuming that the contribution of the polymer part to heat conduction in E-50 is equal to that in E-0, the thermal conductivity of E-50 was calculated to be higher than that of E-0 by only 2.3%. Contrary to this prediction, the measured thermal conductivity of E-50 was a factor of 1.24 greater than that of E-0. Such a difference between the measurement and calculation results might be due to the parts of the deposited Ag layer that did not participate in the electrical percolation network but still contributed to heat conduction by being in contact with nylon 66 nanofibers.

A thermal conductivity increase from N-0 to N-50, which was contributed to by the addition of the Ag layers, was more significant than that from E-0 to E-50 due to the porous morphology of the Ag layers in E-50. Additionally, for E-0 and E-50, the air cavities formed by the porous structure, hinder heat transfer through the nanofibrous mats. Despite the substantial increase in thermal conductivity by the Ag layers in N-50 and the adverse effect of the porous structure in E-50, the thermal conductivity of N-50 was still lower than that of E-0. Such a thermal conductivity enhancement in E-0 and E-50 is mainly attributed to the increase in thermal conductivity achieved through electrospinning, which increased the crystallinity of electrospun NFs. Especially, the electrospinning process may also align molecular chains in NFs along the fiber axes [62].

### 3.4. Mechanical Properties of the Composites

To investigate the mechanical properties of the composites, bending and tensile tests were conducted for E-50 and N-50. Through bending tests of 3000 cycles, good mechanical performance including flexibility and attachment of the interfaces between the Ag and the nylon 66 layers was exhibited, as shown in Figure 6a,b. When E-50 was folded in half and released, it recovered to its original shape. Moreover, there occurred no structural damage, e.g., cracks, except for creases caused by the folding test, as shown in Figure 6c. However, the stress-strain curve of E-50 in Figure 6d shows that the multi-layered structures were fractured one-by-one during the tensile test. It was thought that the merged film may be resistant to bending forces, but disintegrates when subjected to tensile ones. The measured elastic modulus and maximum stress of E-50 are 1085 MPa and 58.5 MPa, respectively, at 0.1322 mm/mm. Compared to the values of nylon 66 nanofibrous mat obtained from a previous study [76], the elastic modulus and maximum stress were enhanced by factors of 23.4 and 12.9, respectively. This is due to the hydrogen bond rupture and subsequent movement of molecular chains, which could increase the crystallinity during the hot-pressing process [47].

### 3.5. EMI SE Effectiveness at X- and K_u_-Bands

Figure 7 shows the measured SE_T_, SE_R_, and SE_A_ values of E-50 and N-50 in the X- and K_u_-bands. The SE values of E-50 were obtained for two different cases: the first one in which the electric fields of the EM waves were parallel and the second one in which they were perpendicular, to the orientation of NFs. The SE of E-50 became higher when the EM waves were parallel with the axial orientation of the NF. This phenomenon was also observed for metallic wire grids, of which the geometries were comparable to the wavelength of the incident EM waves when the direction of the grid was parallel to the polarization of the EM waves [77,78]. Additionally, all the SE_T_ values satisfied the commercial requirement of 30 dB, as shown in Figure 7a [79,80,81]. In particular, due to the porous structure, the SE_T_ of E-50 with a thickness of 100 μm was as high as 60.6 dB in addition to its high thermal conductivity and low density of 0.893 g/cm^3^. Generally, the EMI SE, SSE, and absolute SE (SSE/t, obtained by dividing SSE to its thickness) are useful criteria for comparing EMI shielding performance because there is a demand for lightweight and thin EMI-shielding materials. The calculated SSE and SSE/t values of E-50 are 67.92 dB cm^3^/g and 6792 dB cm^2^/g, respectively. Despite the thinness of the prepared EMI shielding composite, the SE value of E-50 is higher than or comparable to the reported values listed in Table 2. Considering the low filler content and the high thermal conductivity, it was certain that the SSE and SSE/t values of E-50 are also sufficiently outstanding. In this nanofibrous and multi-layered structure, there may exist two additional shielding mechanisms that contribute to absorption: inter-fiber [32] and inter-layer [82] multiple reflections. The inter-fiber multiple reflections mostly occur inside each of the four Ag layers on the nanofibrous mats. Because of the large effective surface areas of NFs, incident EM waves are reflected multiple times from the metal surfaces, possibly resulting in absorption of the EM waves and corresponding heat dissipation. In contrast, the inter-layer multiple reflections arise among the spatially separated Ag layers. As shown in Figure 7b, the SE_A_ value of E-50 is slightly lower than that of N-50 below 15 GHz but can be considered comparable to that of N-50 over the entire range of the X- and Ku-bands. Although the electrical conductivity of N-50 is much higher than that of E-50, the contribution of its high electrical conductivity to SE_A_ was insignificant, as shown in Appendix A. The ratios of SE_A_ to SE_T_, obtained from the electrical conductivity of E-50 and N-50, were not different from each other despite the large difference in their electrical conductivities. The inter-fiber multiple reflections among the NFs would not play a dominant role for the EMI shielding of E-50 considering that the electrospun NFs were well aligned and that a large amount of the empty space between the NFs disappeared when the NFs were hot-pressed as shown in Appendix A.

To investigate the contribution of inter-layer multiple reflections to total absorption, EMI SE values of N-50 were numerically calculated using the commercial finite element software COMSOL (Stockholm, Sweden). The numerical calculation was conducted in a frequency domain based on the following Maxwell equation:(5)$$\nabla \times \frac{1}{\mu_{r}}\left(\nabla \times E\right)=k_{0}^{2}\left(\varepsilon_{r}-\frac{j\sigma }{\omega \varepsilon_{0}}\right)E.$$
In the above equation, μ_r_ is the relative permeability, **E** is the electric field, ε*_r_* is the relative permittivity, σ is the electrical conductivity, $\varepsilon_{0}$ is the vacuum permittivity, and *k*_0_ is the wavenumber of free space. Here, $k_{0}$ is defined as
(6)$$k_{0}=\omega \sqrt{\varepsilon_{0}\mu_{0}}=\frac{\omega }{c},$$
where *c* is the speed of light. To simulate the rectangular waveguides for the X- and K_u_-bands, the boundaries of the waveguides were set to be perfect electrical conductors (e.g., n×E=0, where **n** is the normal vector to the boundary). Additionally, the measured electrical conductivities of the Ag layers and the other material properties were selected from previous studies [61,88,89,90]. The 3D model was spatially discretized into 3329 meshes. More and smaller meshes were allocated to the lengthwise central region of the waveguides, where the samples were located, using the “User defined” mesh type. In contrast, the “Normal” mesh type was applied for the other region (see Appendix A). A workstation with 12 2.5 GHz Intel processor cores and 128 GB of RAM was employed, and the computation time was <10 min for each simulation case. To confirm the numerical calculation results, analytical calculations were also performed for N-50. The detailed information about the analytical calculations is described in the Appendix A.

Figure 8a shows the numerically and analytically calculated SE_T_, SE_A_, and SE_R_ values of N-50. All the numerically calculated results are discontinuous at 12.4 GHz because different waveguides were applied below and above the frequency, which are the X- and K_u_-bands, respectively. The calculated EMI SE values from both methods are similar and both higher than the measured EMI SE values. This might be due to imperfections in the Ag layers in N-50 (such as pores) and its non-uniform thickness, which would occur during the hot pressing process, as shown in Appendix A. When the thickness of the first Ag layer is assumed to be 30 and 50 nm, approximately 2.2 and 1.3% of an EM wave is expected to penetrate the first Ag layer. Such a penetrated wave was mostly absorbed in the composite by multiple reflections among the four Ag layers. More specifically, because both calculations do not involve multiple inter-fiber reflections, the inter-layer multiple reflections of N-50 were obtained from the difference between the total absorption SE (SE_A, total_) and penetration loss (SE_A, penetration loss_). The penetration loss is attenuation that occurs when EM waves pass through a material without reflection. According to the analytical calculation, the penetration loss in N-50 is estimated to be approximately 1 dB. Because the penetration loss is less than 15 dB [91], the multiple inter-layer reflections must account for a significant portion of the SE of N-50. Specifically, 96.7–97.2% of the absorption in N-50 originates from the multiple inter-layer reflections. The absorption mechanism of N-50 is mainly obtained due to the multiple inter-layer reflections in its structure, as shown in Figure 8b. Therefore, inter-layer multiple reflections play an important role in the increase in SE_A_, which accordingly contributes to the increase in SE_T_ of N-50. Similarly, it is expected that the EM waves are absorbed largely by the multiple inter-layer reflections in E-50. These numerical and analytical calculations justify the conclusion that the absorption-dominant SE was realized as a result of the multiple reflections.

## 4. Conclusions

To fabricate the highly thermally-conducting absorption-dominant EMI shield, Ag-deposited electrospun nylon 66 layers were stacked and subsequently hot-pressed. The thermal conductivity of the shield was gradually increased during the fabrication processes. The thermal conductivity of E-0 was increased by a factor of 13.4 from that of the bulk because the crystallinity increased during the fabrication processes and the molecular chains in the nylon 66 NF may be stretched and tightly aligned. The Ag-deposited electrospun mat possessed different electrical properties and SE, depending on the angle between the polarized direction of the incident EM wave and the alignment direction of the nylon 66 NFs. The electrospinning process increased the thermal conductivity, but at the same time reduced the electrical conductivity due to the porous morphology of the Ag layer. Although the lower electrical conductivity results in a slight decrease in EMI SE, much higher thermal conductivity was achieved through the electrospinning process. In addition, numerical and analytical calculations verified that the multi-layered structure induces absorption-dominant EMI shielding by multiple inter-layer reflections. In contrast, multiple inter-fiber reflections were insignificant to absorption SE.

## Figures and Tables

**Figure 1 polymers-12-01805-f001:**
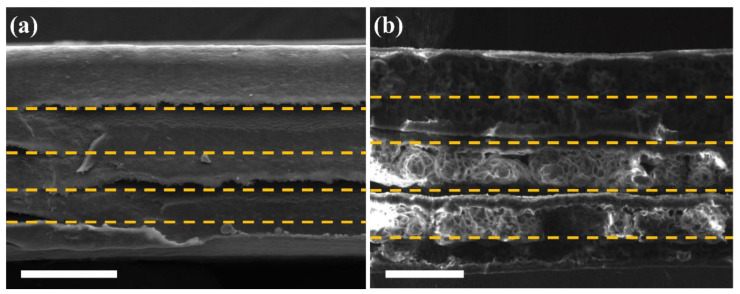
Cross-section view SEM images of the prepared composites including Ag layers such as (**a**) E-50 and (**b**) N-50. The yellow dashed lines indicate interfaces between the four Ag-deposited layers and the electrospun mat or the non-porous film. The multiple layers of N-50 were separated when they were cut for cross-sectional SEM imaging. The scale bars are 50 μm in both (**a**,**b**).

**Figure 2 polymers-12-01805-f002:**
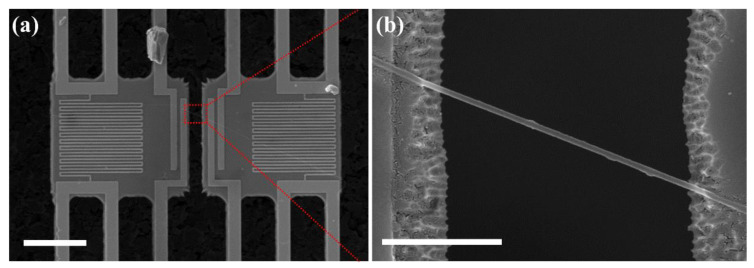
SEM images of the individual electrospun nylon 66 NF assembled on the suspended micro-device. (**a**) The central region of the suspended micro-device, where the suspended membranes and the NF are located. The scale bar is 20 μm. (**b**) Enlarged SEM image of the part of the NF that bridges the two suspended membranes. The scale bar indicates 2 μm.

**Figure 3 polymers-12-01805-f003:**
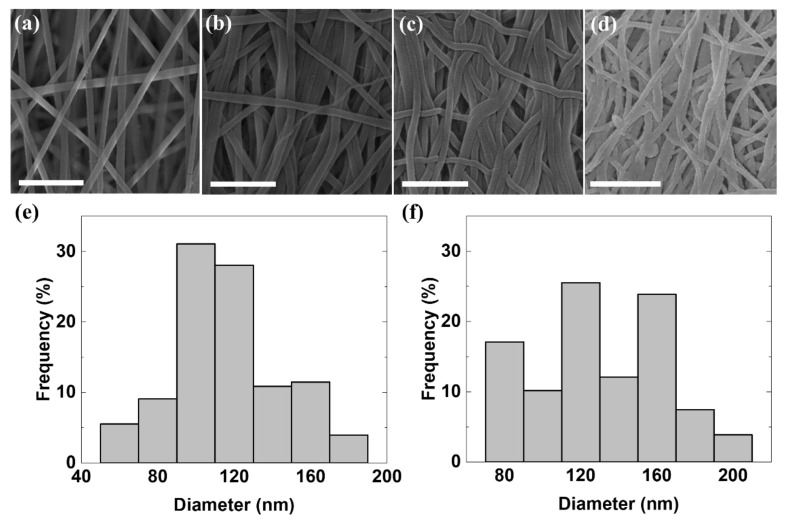
SEM images, which were taken in order of the fabrication process of E-50, of (**a**) As-spun NFs, (**b**) Ethanol-treated NFs, (**c**) An ethanol-treated and then hot-pressed mat, and (**d**) An Ag-deposited hot-pressed mat and corresponding diameter distributions of (**e**) As-spun NFs and (**f**) Ethanol-treated and hot-pressed mat. The ethanol treatment was applied to the as-spun mats before the hot-pressing process to increase adhesion between the NFs. All the scale bars indicate 1 μm. The average diameter and the diameter distribution were measured from five different SEM images of the mats using Image J software (National Institutes of Health, Bethesda, MD, USA). The average diameters of the as-spun and ethanol-treated and hot-pressed mats are 116.0 nm and 130.7 nm, respectively. It was thought that the average diameter of the NFs slightly increased during the fabrication processes.

**Figure 4 polymers-12-01805-f004:**
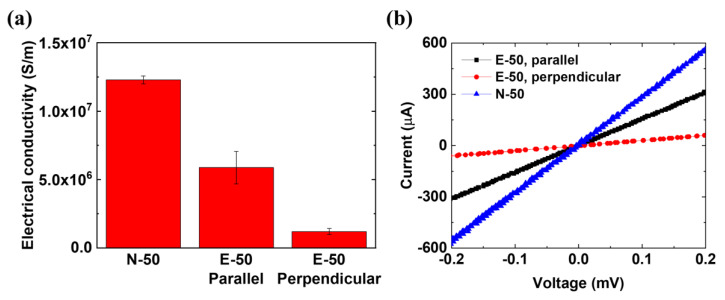
Electrical conductivity of the composites measured using the four-point probe method at room temperature. (**a**) Electrical conductivity values of a single 50-nm-thick Ag layer in N-50 and E-50. The electrical conductivity values of E-50 were obtained for two different directions, which are parallel and perpendicular to the nylon 66 NF alignment. (**b**) Corresponding I–V curves for the electrical conductivity measurements. The electrical conductivity of N-50 is significantly higher than those of the other samples because of the formation of well-connected electrical paths in the deposited Ag layer. As expected, there is no observed anisotropic characteristic depending on the direction for the electrical conductivity of N-50. On the other hand, the electrical conductivity of E-50 is relatively low because of the nanofibrous structure, and the conductivity in the parallel direction is higher than that in the perpendicular direction by a factor of 4.91.

**Figure 5 polymers-12-01805-f005:**
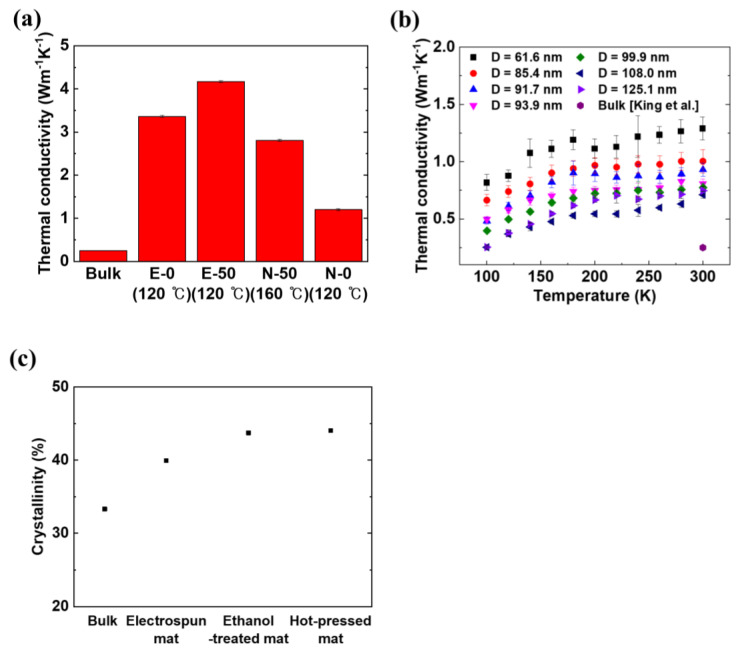
Measured thermal conductivity of mats (E-0 and E-50), films (N-0 and N-50), and electrospun individual NFs and degree of crystallinity of the electrospun mats according to the fabrication stage. (**a**) Thermal conductivity values of four different samples (E-0, E-50, N-50, and N-0) measured at room temperature to clarify the contributions of nanofibrous nylon 66 and deposited Ag to the thermal conductivity. The temperature in parentheses represents the hot-pressing temperature. The thermal conductivity of the mats was increased by the electrospinning and hot-pressing processes, which enhance the alignment of the molecular chains and the crystallinity in the mats. The thermal conductivity value of bulk nylon 66 was taken from previous research [61]. The error bars are not sufficiently large to be seen. (**b**) Measured thermal conductivity of individual electrospun nylon 66 NFs with different diameters, as a function of temperature (100–300 K). The thermal conductivity increases with smaller NF diameter and is 3- to 5-fold higher than that of bulk nylon 66 at room temperature. (**c**) Degree of crystallinity of as-spun, ethanol-treated, and ethanol-treated-hot-pressed (120 °C) electrospun mats, as well as that of bulk nylon 66, which was calculated using the density of a nylon 66 pellet [66]. The degree of crystallinity gradually increases during the fabrication stage, especially during the electrospinning process.

**Figure 6 polymers-12-01805-f006:**
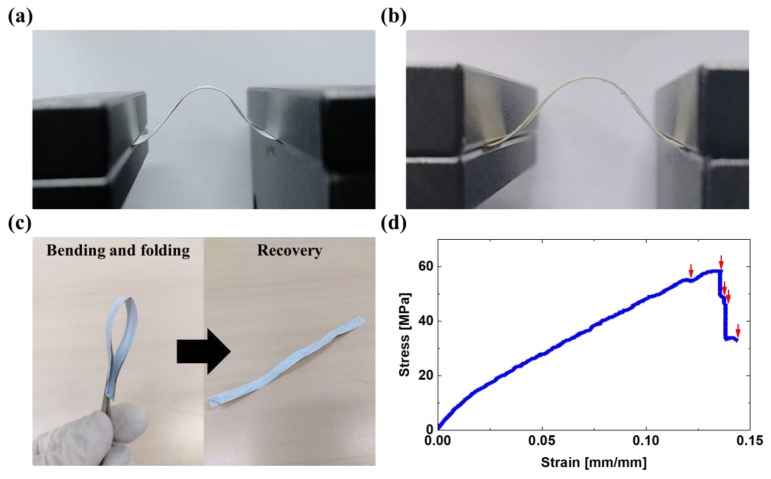
Tests to confirm the mechanical performance of the EMI shielding composites such as E-50 and N-50. Digital images of the cyclic bending tests for (**a**) E-50 and (**b**) N-50 using a custom motorized stage. (**c**) Digital image of the flexibility test of E-50. After the bending and flexibility tests of E-50 and N-50, any detachment at the interfaces between Ag and nylon 66 layers was not observed. (**d**) Stress-strain curve of E-50. The red arrows indicate the strain values at which each layer was fractured one by one.

**Figure 7 polymers-12-01805-f007:**
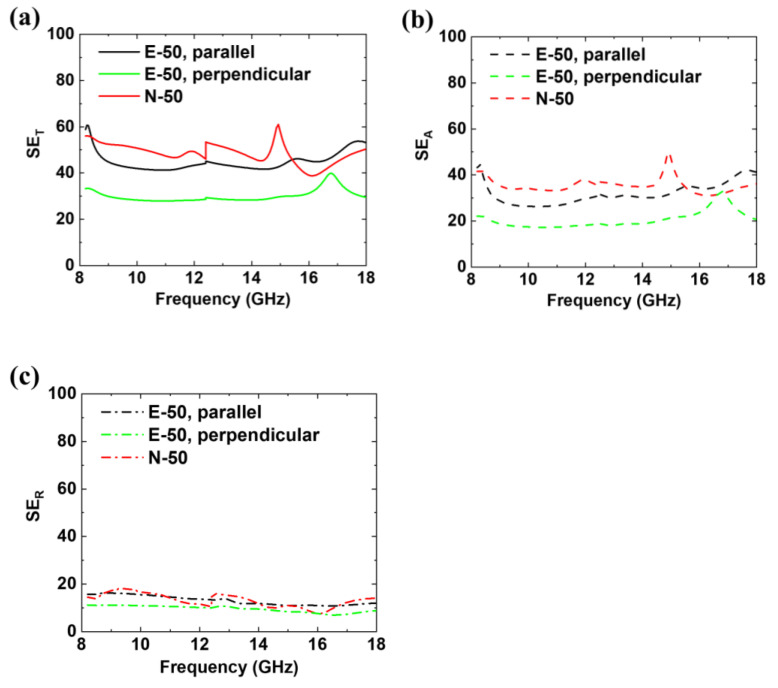
Measured EMI SE of E-50 in the direction parallel (black) and perpendicular (green) to the axial orientation of the NFs and N-50 (red) at the X- and K_u_-bands. The discontinuity at 12.4 GHz originates from the different waveguides for measurements at the X- and K_u_-bands. (**a**) Total EMI SE (SE_T_, continuous line) was calculated by the sum of (**b**) Absorption (SE_A_, dashed line) and (**c**) Reflection (SE_R_, dash-dotted line), which were obtained from measured S-parameters (S_11_ and S_21_) using a network analyzer. All the values in SE_T_ are greater than 30 dB, which means that all the samples meet the commercial requirement for an EMI shield, and the portion of the SE_A_ accounts for much more than that of SE_R_ for the total EMI SE. In addition, the anisotropic characteristic of E-50 is also exhibited by its electrical conductivity.

**Figure 8 polymers-12-01805-f008:**
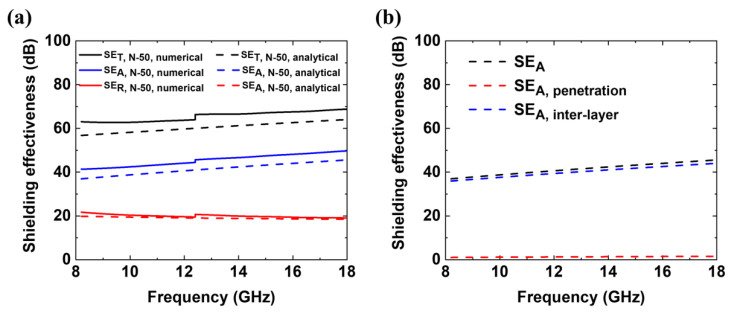
Numerically (continuous line) and analytically (dashed line) calculated shielding effectiveness of N-50 at the X- and K_u_-bands. (**a**) Numerically and analytically calculated SE_T_ (black), SE_A_ (blue), and SE_R_ (red) of N-50 agree well with each other. (**b**) Analytically calculated total SE_A_ (black), which is composed of the multiple inter-layer reflections (blue) and the penetration loss (red), at the X- and Ku-bands. Most of the total absorption is due to the inter-layer multiple reflections mechanism.

**Table 1 polymers-12-01805-t001:** Fabrication of EMI Shielding mat (E-50) with deposited Ag layers. The other samples (E-0, N-0, and N-50) were also fabricated to compare the EMI SE and thermal conductivity.

Sample	Polymer Processing	Ag Layer Thickness (nm)	Hot-Pressing Temperature (°C)/Pressure (MPa)
E-0	Electrospinning	None	120/10
E-50	Electrospinning	50/50/50/50	120/10
N-0	Doctor blading	None	120/10
N-50	Doctor blading	50/50/50/50	160/10

**Table 2 polymers-12-01805-t002:** Comparison of EMI shielding at X and/or Ku-bands and thermal performance of E-50 with the reported composites. The abbreviations, t and k, are the thickness and the thermal conductivity of the composites. Notably, the highest SE values at the specific frequency range were referred.

Filler (Content)	Matrix	Frequency Range (GHz)	t (cm)	SE (dB)	SSE (dB cm^3^/g)	SSE/t (dB cm^2^/g)	k (Wm^−1^K^−1^)	Ref
Ag NWs (4.5 wt%)	PI foam	8–12	0.5	2.85	233	465	-	[83]
Ag NWs (28.6 wt%)	Waterborne PU	8.2–12.4	0.23	64	1422	6183	-	[84]
Ag NWs (–)	Wood-derived carbon with N-doped graphene	8.2–12.4	0.15	44.2	340	2266	0.141	[85]
Ag NPs (–)	Crosslinking PAN NFs	8–12	0.0035	83.7	39.39	11,254	-	[34]
Ag NWs (~67 wt%)	Carbon sponge	8.2–18	0.1	37.9	9921	99,214	-	[86]
Ag NWs (0.1 wt%)	Ti_3_C_2_T_x_ MXene	8.2–12.4	0.00169	42	28	16,724	-	[87]
GNSs (5 wt%) and CINAP (15 wt%)	Cyanate ester	8.2–12.4	0.35	55	-	-	4.13	[38]
Cu NWs (7.2 wt%)	Graphene aerogel	8.2–12.4	0.2	47	-	-	0.51	[25]
Ag thin film (2.4 wt%)	Nylon 66 NFs	8.2–18	0.01	60.63	67.92	6792	4.17	This work

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
