# Peer review of "Electromagnetic Interference Shield of Highly Thermal-Conducting, Light-Weight, and Flexible Electrospun Nylon 66 Nanofiber-Silver Multi-Layer Film"

_polymers, 2020, doi:10.3390/polym12081805_

Round 1

Reviewer 1 Report

The manuscript “Electromagnetic interference shield of highly thermal-conducting, light-weight, and flexible electrospun nylon 66 nanofiber-silver multi-layer film” reports four types of electromagnetic wave shielding film of different structure based on nylon 66 nanofiber. The electromagnetic wave shielding film is an area attracting attention and it can be said that it is a unique research in that it fabricates and compares the film with different laminated structure and component. However, on the other hand, I think that some fundamental improvements are necessary for the structure and description of this manuscript before publishing in Polymers.

  1. The authors should provide the SEM images of the obtained multi-layer film with different structure, so as to readers can better understand this article.
  2. All figures in this manuscript should be optimized for reader-friendly.
  3. In “2.1 materials”, acronym ‘MW’ should be previously defined.
  4. Authors mentioned that “To reduce the randomness of the NFs orientation in the mat, the rotation speed of the drum collector was set to 1600 rpm.” Why? Does a higher or lower rotation speed generate a more efficient reduction in randomness.
  5. The title illustrates that the films possesses flexibility, but the manuscript does not discuss this characteristic of flexibility. Please authors provide some test or digital images about flexibility. There are some references that may help. Such as “Multi-layered graphene-Fe3O4/poly (vinylidene fluoride) hybrid composite films for high-efficient electromagnetic shielding”.
  6. The electrical conductivity of film prepared in this article is very high. For high conductivity materials, reflection is always the leading mechanism. Why this article can achieve absorption-dominant EMI shielding in such high conductivity.
  7. In the “Introduction”, the relationship between EMI shielding and thermal conductivity should be highlighted. The authors should cite some references, such as “Synergistic effect of graphene nanosheets and carbonyl iron-nickel alloy hybrid filler on electromagnetic interference shielding and thermal conductivity of cyanate eater composites”.

Reviewer 2 Report

In this study, the author presents a layer-structured metal-polymer composite film consisted of electro-spun nylon 66 nanofibers and thin silver layers. On the X- and K u -bands, the EMI shielding effectiveness (SE) of the composite was up to 45 dB on average. By numerical and analytical calculations, it is suggested that the energy of EM waves is predominantly absorbed by inter-layer reflections. But in this paper, there are still some shortcomings that need to be further modified.

  1. The abstract is not concise, and it is suggested that the author should rewrite the abstract. For example, the background introduction is too long.
  2. The innovation of this study presented by the author should be further elaborated and the author should compare its SSE and SSE/t with other similar research results.
  3. What is the change in fiber diameter of ethanol-treated and hot-pressed NFs? The author should supplement the corresponding fiber diameter distribution map.
  4. “To fabricate the EMI shielding composites, 50 nm-thick Ag layers were deposited on hot-pressed electro-spun mats and non-porous films, respectively, using an electron beam evaporator (REP5004, SNTek, Korea) with a deposition rate of 0.7 Å/s. The five Ag-deposited electro-spun mats were piled up along the direction of the NFs and hot-pressed at 120 °C afterward to form a single, merged electro-spun mat, denoted by E-50.” To this statement, why the thickness of Ag is controlled at 50nm and what effect does the different Ag coating thickness have on EMI shielding? “The five Ag-deposited electro-spun mats” are described as 5 here, but the number of Ag deposited layers in Table1 is 4, the author needs to explain it.
  5. 50 nm-thick Ag layers were deposited on hot-pressed electro-spun mats and non-porous films, how fast is the combination of the two? The author should supplement the description in this part.
  6. The error bar for conductivity and thermal conductivity should be added in Fig. 3 and Fig. 4.
  7. In the Introduction section, more references (Compos. Pt. A-Appl. Sci. Manuf. 2020, 128. et al.) published in recent years should be cited.
  8. The references format should be rechecked according to the requirement of Polymers. Please revise the references of 2, 10, 11, 12, 28, 54, 57, 62.
  9. Some language and grammar errors need to be corrected before publication.

Reviewer 3 Report

The main theme in this paper seems to be the effect of porous structures in their multi-layered composites (nanofibrous mat) when referring to their sample preparation table. However, the main discussions on their thermal conductivity and EMI did not pertain much to the presence of porous structure. The authors must provide much detailed discussions on the effect of porous structures especially for thermal conductivity section.

  1. Which sample was examined by SEM for Fig 2? Indicate that which sample was chosen from Table 1. Also show all the SEM images for each composite listed in Table 1.

  1. Provide pore size distributions for E-50 and E-0.

  1. Please justify the reason for using electrospun for creating porous structures.

  1. Please explain the effect of pore structures on EMI and thermal conductivity in comparison to N-50 composites.

  1. In figure 4-(a), the thermal conductivity of E-0 and E-50 was higher than N-50 and N-0 even with the presence of pores, which are highly thermally insulating. Please provide further explanations for this result. How significant is the effect of crystallinity against the presence of pores for improving thermal conductivity?

  1. In line 210 on page 6, please explain how electrical conductivity is associated with the variation of the electron mean fee path and provide relevant references for readers.

  1. In abstract, the authors claimed that “Because the absorbed EM energy is dissipated as heat, the thermal conductivity of absorption-dominant EMI shields is highly significant”. Please provide further discussion on how thermal conductivity varied with the EMI mechanism when it is absorption dominant or reflection dominant. Please provide detailed theory and relevant references for readers.

Reviewer 4 Report

The presented research work discusses the ability of EM shielding using nylon electrospun nanofibers embedded with silver. Overall, the analysis is promising and worth to be published but there are minor points have to be addressed first as follows:

1- The I-V curves should be presented before the bar diagram of the conductivity values.

2- Porosity calculations should be presented to show the value of non-porous three layers compared to the non-stacked layers.

3- Nylon is used due to its mechanical properties, in addition to the electrical ones. So, stress-strain curves of the different used layers have to be presented with analyzing the corresponding properties such as elasticity modulus, maximum elastic strain, breaking point stress and strain.

4- Comparison between shielding effectiveness of the synthesized nylon mats and other commercial/literature shields should be discussed.

Round 2

Reviewer 1 Report

The authors have made detailed modifications to the article. Hence I recommended acceptance to this manuscript.

Reviewer 3 Report

Now it can be accepted for publication